# Nitric Oxide, Oxidative Stress and *Streptococcus mutans* and *Lactobacillus* Bacterial Loads in Saliva during the Different Stages of Pregnancy: A Longitudinal Study

**DOI:** 10.3390/ijerph18179330

**Published:** 2021-09-03

**Authors:** Madhu Wagle, Purusotam Basnet, Åse Vårtun, Ganesh Acharya

**Affiliations:** 1Women’s Health and Perinatology Research Group, Department of Clinical Medicine, Faculty of Health Sciences, UiT—The Arctic University of Norway, 9037 Tromsø, Norway; ase.vartun@uit.no (Å.V.); ganesh.acharya@ki.se (G.A.); 2Department of Obstetrics and Gynecology, University Hospital of North Norway, 9038 Tromsø, Norway; 3Department of Clinical Science, Intervention and Technology, Division of Obstetrics and Gynaecology, Karolinska Institutet, 141 86 Stockholm, Sweden

**Keywords:** antioxidant capacity (AC), *Lactobacillus*, malondialdehyde (MDA), nitric oxide (NO), oral health, oxidative stress (OS), pregnancy, saliva, *Streptococcus mutans*

## Abstract

Hormonal changes associated with pregnancy promote oral bacterial growth, which may affect salivary nitric oxide (NO) levels, oxidative stress (OS), and antioxidant capacity (AC). We hypothesized that caries-related bacterial load, NO level, and OS in the saliva change with advancing gestation. The aim of this study was to investigate longitudinal changes in salivary NO, OS, and AC during pregnancy and correlate them with *Streptococcus mutans* (*SM*) and *Lactobacillus* (*LB*) colonization at different stages of pregnancy. We assessed NO level by Griess method, OS by measuring malondialdehyde (MDA), AC by ABTS radicals and bacterial load by culturing *SM* and *LB* in the saliva of pregnant women (*n* = 96) and compared with non-pregnant women (*n* = 50) as well as between different stages of pregnancy. Compared with non-pregnant women, NO was 77% higher (4.73 ± 2.87 vs. 2.67 ± 1.55 µM; *p* < 0.001), MDA was 13% higher (0.96 ± 0.27 vs. 0.85 ± 0.22 nM; *p* = 0.0055), and AC was 34% lower (60.35 ± 14.33 vs. 80.82 ± 11.60%; *p* < 0.001) in the late third trimester. NO increased with advancing gestation, but AC and OS did not change significantly during pregnancy. *SM* were more abundant in pregnant women compared with non-pregnant (*p* = 0.0012). Pregnancy appears to have an adverse impact on oral health emphasizing the importance optimal oral healthcare during pregnancy.

## 1. Introduction

Saliva plays a vital role in the digestive tract contributing to the maintenance, preservation, lubrication, protection, and healing of intra-oral hard and soft tissues. Saliva is widely used as a screening and diagnostic tool because of its simple non-invasive collection process and the presence of several biomarkers [1,2,3]. Normal pregnancy is associated with altered flow, composition, and pH of saliva, which promotes bacterial growth and predisposes women to poor oral health [4]. Furthermore, physiological endocrine, cardiovascular, immunological, and metabolic adaptations of pregnancy may also contribute to increased colonization of oral cavity by pathogenic microbes leading to inflammatory response and oxidative stress. Nitric oxide (NO) has a key role in pregnancy as a regulator of both maternal and fetal homeostasis [5], and its levels in the saliva are likely to reflect local conditions of the oral cavity. However, although there are a limited number of cross-sectional studies measuring NO levels in maternal serum during pregnancy [6,7], no study to our knowledge has performed NO measurements in the saliva of pregnant women longitudinally. Saliva also plays an important role in the immunological and enzymatic defense mechanisms against certain microorganisms’ antioxidant systems, and human body’s oxidative stress (OS) levels are expressed in saliva [8]. A recent systematic review on maternal oral microflora has reported that the pregnancy is associated with abundance of micro-organisms compared to non-pregnant state [9]. In a cross-sectional study, we have previously demonstrated an increased level of OS in the saliva of pregnant women compared to non-pregnant in association with increased colonization of oral cavity by dental caries-related pathogen, *Streptococcus mutans* (*SM*) [10]. Increased levels of NO and OS have been measured in the blood samples of pregnant women compared to healthy non-pregnant women [6,11]. Therefore, longitudinal measurements of NO and OS levels in the saliva of pregnant women, together with the assessment of pathogenic bacterial load could help to elucidate the link between OS and oral health during the course of normal pregnancy. We hypothesized that the colonization by caries-related bacteria, OS level, and antioxidant capacity (AC) change during pregnancy with advancing gestational age. We aimed to investigate associations of oral pathogenic bacterial load represented by *SM* and *Lactobacillus* (*LB*), with salivary NO and OS levels during the progression of pregnancy.

## 2. Materials and Methods

### 2.1. Study Design

This single center observational cohort with comparison group was part of a longitudinal study on oral health in pregnancy conducted at UiT-The Arctic University of Norway and University Hospital of North Norway, Tromsø, Norway. Saliva samples were collected consecutively from healthy pregnant women four times during gestation starting at 18–20 weeks to term. The samples were divided into 4 groups by gestational age; early second trimester (18–20 weeks), late second trimester (24–26 weeks), early third trimester (30–32 weeks), and late third trimester (36–40 weeks). The comparison group comprised of healthy non-pregnant women of reproductive age without history of any chronic illness or recent disease. These were recruited among women working at the UiT-The Arctic University of Norway or the University Hospital of North Norway, Tromsø. Pregnant women were approached and recruited to the study when they attended the hospital for routine ultrasound screening at 18–20 weeks of gestation. General inclusion criteria were: (1) age over 18 years, (2) low risk singleton pregnancy, (3) women without any previous history of pregnancy-associated complications, such as preeclampsia, preterm birth, or gestational diabetes, and (4) absence of any pre-existing medical condition that may have an impact on the course and outcome of pregnancy. Those pregnant women who were unable to communicate in Norwegian or English, and those who have been diagnosed with a fetal chromosomal or structural anomaly and did not plan to continue their pregnancy, were excluded. A history of any acute or chronic illness requiring regular medical treatment excluded participation for the non-pregnant women. Pregnant study participants were followed from mid-gestation until childbirth, and the outcome of pregnancy was recorded.

All participants were informed about the study in advance and a written consent was obtained from all the participants. The study was approved by the Regional Committee for Medical and Health Research Ethics-North Norway (Ref No: 2012/633/REK nord).

### 2.2. Collection of Samples

For both pregnant and non-pregnant groups, saliva samples were collected during day-time between 9:00 and 15:00 h with the same method as described previously [10]. Women were not eating or brushing their teeth for approximately 1.0 to 1.5 h before sampling. In brief, the participants were asked chew paraffin wax pellets for 5 min to stimulate saliva production which was then collected by expectorating in sterile disposable cups.

For the assessment of caries related bacterial load in the saliva, *SM* and *LB* were cultured and colony forming units (CFU) were evaluated by a single investigator (MW). For the measurement of NO and OS levels, 1.8 mL of saliva was collected in two separate cryo-tube vials and stored at −70 °C until samples were analyzed by a single investigator (PB). Samples were put in a refrigerator at 4 °C for one day before the analyses were performed. All samples were left for 2 h on the working table before the analysis to defrost and bring to room temperature. Sample containing tubes were centrifuged at 10,000× *g* for 10 min to remove cell debris and supernatant was collected for further analysis. Storing-procedures and laboratory analyses were followed according to the instructions by the kit supplier.

### 2.3. Chemicals

All chemical used were of analytical grade. 2,2′-azino bis(3-ethylbenzothiazoline)-6-sulfonic acid diammonium salt (ABTS), vitamin C (ascorbic acid), acetic acid, and sodium nitrite were purchased from Sigma-Aldrich, Chemie GmbH, Steinheim, Germany. Potassium peroxodisulphate was a product from Merck KGaA, Darmstadt, Germany. Griess reagent was prepared from 1% sulfanilamide, 0.1% naphthylethylene diamine dihydrochloride, and 2.5% phosphoric acid, all products were from Sigma-Aldrich Norway AS, Oslo.

### 2.4. Measurement of NO Levels

Because NO is highly labile, measurement of the relatively stable metabolite, nitrate and nitrite (NO_x_), is employed as an index of NO production and, also, as a marker of NOS enzyme activity [12]. The levels of NO in saliva were expressed by measuring nitrite quantitatively using the Griess method spectrophotometrically with modification [13]. Griess reagent was prepared as 1% sulfanilamide and 0.1% N-naphthylethylene diamine dichloride in 5% orthophosphoric acid (*v*/*v*). This reagent reacts with nitrite and produces a purple azo dye end-product, which is measured spectrophotometrically with a maximum absorbance at 546 nm. Triplicate samples of saliva (10 μL) were transferred to a tube containing 290 μL of distilled water, and a 300 μL of Griess reagent was added to each tube. After mixing thoroughly and allowing to react at dark for 30 min, the changed color was measured. The quantitative expression of NO in saliva is taken from the analysis of triplicates of sodium nitrite (NaNO_2_) at concentrations of 25, 20, 15, 10, 5, 2, 1, and 0 μM as final concentrations and with the help of standard curve and regression equation (R^2^ = 0.9995 and y = 0369x − 0.0035). The intra-assay coefficient of variation was 4.89%

### 2.5. Measurement of Antioxidant Capacity (AC)

Antioxidant capacity (AC) in the saliva was assessed by measuring free radical scavenging activity by 2,2′-azino-bis(3-ethylbenzothiazoline)-6-sulfonic acid diammonium salt (ABTS) methods with appropriate modifications [10,13]. In brief, ABTS free radicals (ABTS^•+^) having dark green color was generated by allowing the reaction equal volume (each 2 mL) of the solutions of ABTS (7.4 mM) and potassium peroxodisulfate (2.6 mM) for 24 h. The reaction mixture was diluted to 100 mL with distilled water as a working solution ABTS^•+^. Optical density (OD) of the working ABTS^•+^solution was approximately 0.5 to 0.6. Supernatant of saliva samples were used for both groups. Reactions were carried out by mixing 450 μL of working solution of ABTS radical and 50 μL supernatant part of saliva followed by incubating for 30 min in darkness. The changes in the green color of ABTS free radicals scavenged by the antioxidants present in saliva fluid was measured for OD using spectrophotometric methods (Agilent Technologies Deutschland GmbH, Waldbronn, Germany) at 731 nm. Higher OD_731_ value represents lower level of TAC. Water soluble vitamin C (Sigma-Aldrich) was used as a standard and AC was quantified as μg/mL vitamin C equivalent level, representing AC with the help of standard curve and regression equation (R^2^ = 0.931 and y = −0.0304x + 0.6134). The intra-assay coefficient of variation was 0.73%.

### 2.6. Measurement of Oxidative Stress

We evaluated OS levels in saliva by measuring MDA content using a commercially available MDA Assay Kit (Sigma-Aldrich, Lipid Peroxidation MDA Assay Kit (MAK085), Darmstadt, Germany) [10]. In brief, a mixture of 100 µL saliva supernatant, 100 µL buffer provided in the kit, and 600 µL thiobarbituric acid (TBA) solution were mixed thoroughly and incubated at 95 °C for 60 min. Of the reaction mixture, after cooling in ice, 150 µL was transferred to a 96 well microplate in duplicates and absorbance was measured fluorometrically (Epoch Microplate, BioTek Instrument, Winooski, VT, USA) by measuring fluorescence intensity (λ_ex_ = 532/λ_em_ = 553). The MDA levels in the saliva were calculated by the MDA standard provided in the kit with the help of standard curve and regression equation (R^2^ = 0.9951 and y = 783.07x + 12.095). The intra-assay coefficient of variation was 6.91%.

### 2.7. Assessment of SM and LB in Saliva

Oral bacterial culture was assessed by a single investigator (MW) by saliva culture and development of bacterial colony forming units (CFU) of two main bacteria, *SM* and *LB*, using commercial kits Dentocult^®^ *LB* (kit for *LB*), and Dentocult^®^ *SM* Strip mutans (kit for *SM*) (Orion Diagnostica Oy, Espoo, Finland) as described previously [10]. Women were requested to chew a paraffin pellet to stimulate the secretion of saliva and promote transfer of *SM* from tooth surfaces into the saliva. A round-tipped test strip supplied in the kits was pressed against the saliva on the woman’s tongue. The strip was placed in the cap of the vail containing culture broth and was recapped in the vail. The vial was loosely capped and incubated at 37 °C and 5% CO_2_ for 48 h. Results were interpreted by scoring as Category 0, 1, 2, and 3 for 0, <10^5^, 10^5^–10^6^, and >10^6^ CFU/mL, respectively, by comparing to the template reader provided in the kits. In case of *LB* culture, the test strip was thoroughly made wet by saliva, fixed in the cap and fitted in the vials containing culture broth. It was then incubated for 4 days at 37 °C and 5% CO_2_. Results were interpreted scoring as Category 0, 1, 2, 3, and 4 for 0, 10^3^, 10^4^, 10^5^, and 10^6^ CFU/mL, respectively, by comparing to the template reader provided in the kits. Results are expressed as the percentage of women in each category based on the development of bacterial CFU. Both Dentocult^®^ *SM* and Dentocult^®^ *LB* have been shown to have good reliability in determining different categories of CFU counts and have a good correlation with standard culture techniques using Agar plates [14,15].

### 2.8. Statistical Analysis

The sample size required for the detection of 15% difference in the OS level between pregnant and non-pregnant women, with 80% power at an alpha of 0.05, was calculated to be at least 38 individuals per group based on mean MDA level and standard deviation (SD) reported in the saliva of 25 healthy female controls in a previous report [16] using an online sample size calculator [17]. As our study on the pregnant women was longitudinal, we recruited more participants to compensate for possible non-attendance or drop-outs at follow-up visits, inadequate sample collection or analysis failure, to ensure that there would be adequate number of samples/measurements for each stage of pregnancy.

Data analysis was performed using IBM SPSS Statistics for Windows, Version 25.0. (IBM Corp, Armonk, NY, USA). Data are presented as mean (SD) as appropriate. Frequency tables were made and comparison between the pregnant and non-pregnant groups was carried out by conducting χ^2^ (chi-squared) test for categorical variables an independent sample t-test for parametric continuous variables. Bonferroni correction was used when multiple comparisons were performed. Comparisons between different stages of pregnancy were made using analysis of variance (ANOVA) and Turkey’s posthoc test was used to find out specific differences between multiple groups when the ANOVA was significant. Associations of laboratory measured saliva parameters (i.e., NO, AC, MDA, and *SM* and *LB* colony forming units) with pregnancy outcomes were analyzed using linear regression. The strength of association between two continuous variables was assessed by Pearson’s correlation coefficient. A *p*-value of <0.05 was considered statistically significant.

## 3. Results

Flow of study participants is shown in Figure 1. Data from a total of 96 pregnant and 50 non-pregnant women were included in the final analysis. The mean ages of the pregnant and non-pregnant groups were 30.81 ± 4.32 and 30.32 ± 6.09 years, respectively. The mean gestational age at delivery was 39.51 ± 1.91 weeks with 80.2% (*n* = 77) women delivering vaginally, whereas 19.8% (*n* = 19) had a caesarean section. Six women delivered preterm (<37 weeks). Mean estimated blood loss during delivery was 498.06 ± 627.11 mL. Similarly, 53.3% of the neonates born were male and 46.7% females, and the mean weight and length of the neonates were 3507.32 ± 631.67 gm and 48.87 ± 7.87 cm, respectively. The median one- and five-minute Apgar scores of the neonates were 9 (range 1 to 10) and 10 (range 3 to 10), respectively.

For the longitudinal evaluation, saliva samples were collected consecutively from 96 healthy singleton pregnant women four times starting at 18–20 weeks to term pregnancy and from 50 healthy non-pregnant women. The samples were divided into four groups by gestational age: early second trimester (18–20 weeks, *n* = 96), late second trimester (24–26 weeks, *n* = 65), early third trimester (30–32.0 weeks, *n* = 56), and late third trimester (36–40 weeks, *n* = 31). The comparison group comprised of healthy non-pregnant women of reproductive age (*n* = 50) without history of any chronic illness or recent disease.

### 3.1. Nitric Oxide (NO) Levels in Saliva

Results for saliva nitrite concentrations of the subjects are shown in Table 1. The average saliva NO equivalent nitrite concentration in the 96 pregnant women measured in the early second trimester was 3.47 ± 2.48 µM, which was 30% higher compared to the group of non-pregnant women (2.67 ± 1.55 µM; *p* = 0.00029). Results showed that NO level continue to increase throughout the pregnancy, attaining peak levels in the end of third trimester (4.73 ± 2.87 µM). The levels of NO equivalent nitrite levels in late third trimester were 36% and 77% higher than in early second trimester of pregnancy (4.73 ± 2.87 µM) and in non-pregnant women (2.67 ± 1.55 µM), respectively. We found no statistically significant association between salivary NO level measured with pregnancy outcomes, such as gestational age at delivery, birthweight, or Apgar score.

### 3.2. Antioxidant capacity (AC) in Saliva

The results of average AC in the saliva of pregnant and non-pregnant women are shown in Table 2. The AC is expressed as the measure of free radical scavenging capacity of the saliva. Higher level of AC shows the saliva containing strong antioxidants. It can be generalized that higher level of AC indicates low level of OS. The average saliva AC levels of pregnant women (*n* = 96) measured at the early second trimester were 60.35 ± 14.33% which is a 25% decrease in free radical scavenging capacity compared to non-pregnant women (80.82 ± 11.60%; *p* < 0.00001). The AC levels did not change significantly throughout pregnancy. AC levels in the saliva of pregnant women (*n* = 96) and non-pregnant women (*n* = 50) were calculated as 16.26 μM and 11.89 μM vitamin C equivalent, respectively, with the regression equation: y = −0.0304x + 0.6134. No statistically significant association was found between AC and pregnancy outcomes.

### 3.3. Oxidative Stress (OS) Levels in Saliva

The results of MDA contents in the group of pregnant and non-pregnant women are shown in Table 3. The OS levels are expressed as the MDA content present in the saliva. The pregnant women had higher levels of MDA in their saliva compared to the non-pregnant women reflecting higher level of OS in pregnant women. The average OS levels, expressed as MDA levels in the saliva of pregnant women in early second trimester gestation (*n* = 96) and non-pregnant women (*n* = 50) were 0.93 nM and 0.85 nM; *p* = 0.008), respectively. OS levels did not change significantly throughout second half of pregnancy. Association between salivary OS level and pregnancy outcomes was not statistically significant.

### 3.4. Oral Bacterial Loads

#### 3.4.1. *Streptococcus mutans* (SM)

The *SM* bacterial loads as compared between the groups of pregnant and non-pregnant women are shown in Table 4. Almost 40% of pregnant women in their early and late third trimester groups developed more than 10^5^.CFU/mL *SM* colonies compared to 22% of non-pregnant women. SM colonies were abundant and significantly higher in saliva of pregnant compared to non-pregnant women (χ^2^ statistic = 5.375; *p* = 0.001). The colony formation was significantly different at different stages of pregnancy (*p* = 0.0024) with the highest proportion (38%) of women being colonized in the early third trimester. However, the degree of SM colonization in the early second trimester was not significantly associated with pregnancy outcomes.

#### 3.4.2. *Lactobacillus* (LB)

The *LB* bacterial loads as compared between the groups of pregnant and non-pregnant women are shown in the Table 5. The *LB* bacterial colonies formation were high and significant during the different stages of pregnancy (*p* = 0.0027). Even though the *LB* bacterial colonies were abundant in the saliva of pregnant women, it was not significantly higher compared to non-pregnant women (χ^2^ statistic = 1.165; *p* = 0.489). Among pregnant population almost 17% had developed 10^6^ CFU/mL *LB* colonies in their late pregnancy stage compared to 4% of non-pregnant women. Majority of non-pregnant showed 10^3^ or less CFU/mL *LB* colonies in their saliva. The degree of *SM* colonization in the early second trimester was not significantly associated with pregnancy outcomes.

## 4. Discussion

In this study, we investigated the changes in NO production in saliva during pregnancy and the levels of NO production were compared with non-pregnant healthy women of reproductive age. We found a significantly higher NO production in the pregnant women compared to the non-pregnant women and NO production also increased with gestational age during normal pregnancy. It is, to our knowledge, the first report of the NO measurements is saliva of pregnant women, however NO production in the blood of pregnant women has been previously reported [6]. Based on some reports evaluating blood samples, NO production is increased in the second trimester and peaks in the third trimester [6,18,19]. In contrast to this, Hata et al. has reported that maternal circulating nitrite level decrease with advancing gestation [20]. Brown et al. [21] and Smarason et al. [22] found no changes in NO production during normal pregnancy compared to non-pregnant women. These studies suggest that the status of NO production in women during normal pregnancy is still controversial and needs further investigation. These discrepancies may derive from methodological variations. Many of the previous studies have relied on the measurement of NO_x_ in the plasma; however, the plasma level is influenced by the clearance, as well as the production of NO metabolites [23].

Saliva also has another very important function in maintaining oral health by regulating the microbial taxa in the oral cavity. Altered oral bacterial milieu and chemical components of saliva have been associated with several diseases [1,2,3,8]. However, only a few studies have investigated the effects of advancing gestational age on oral bacterial load together with OS. In this study, we have measured NO levels in the saliva at different gestational stages together with the OS and oral bacterial load of cries-related pathogens. It is remarkable to find that pregnancy adversely affects oral health by promoting abnormal bacterial growth and increasing NO and OS levels which are expressed in the saliva. It could be that the NO production is a defense mechanism of the body against deteriorating oral hygiene [24]. In 1999, Silva Mendez et al. reported that the nitrite present in saliva could influence growth and survival of cariogenic bacteria [25].

Oral cavity harbors over 700 microbial taxa [26,27] and some microbial species are shown to cause intrauterine infection without being present in the urogenital tract [28,29,30]. A study by Aagaard et al. has shown that microbes found in the term placenta are more similar to those found in the oral cavity rather than vaginal microbes [31]. Pregnancy is known to increase the risk for the development of dental caries compared to the non-pregnant women [32]. *SM* and *LB* are considered as the causative bacteria for developing dental caries [32,33]. Among these, *SM* is reported to be found only in human oral cavity [34]. With this background, we measured *SM* and *LB* load in the saliva of pregnant women at the different stages of physiological pregnancy and compared this with that of the non-pregnant healthy women of similar age. Among healthy non-pregnant women, 20 out of 50 (40%) did not show the presence of *SM* in their saliva. Whereas in the pregnant population, only 27 (10.5%) out of 248 total tested cases at various stages of pregnancies, did not demonstrate the presence of *SM* in the saliva. These results clearly show that a significantly higher number of pregnant women carry *SM* bacterial load in their oral cavity (almost 89.5%) compared to non-pregnant women (60%). Our results partly support the finding of previous studies reporting all pregnant women (100%) to have abundant *SM* colonies in their saliva [32,35]. *SM* bacterial counts at different stages of pregnancy were similar to each other. In general, both *SM* and *LB* bacterial loads were found higher in pregnant women’s saliva compared to non-pregnant women, however, the difference was statistically significant only for *SM.*

It is commonly accepted that OS has a part in the initiation and progression of most oral diseases [36]. However, how the pregnancy associated changes in oral microbial load (especially that of pathogenic bacteria) influence salivary OS and AC has not been elucidated. We found lower levels of AC and higher levels of OS in pregnant women. Nevertheless, the oral bacterial counts of *SM* and *LB* did not significantly correlate with AC or OS neither in pregnant nor in non-pregnant groups. Therefore, it still remains unclear if it is the decreased levels of AC and increased levels of OS that creates a favorable condition for oral bacterial growth in pregnant women or is it the increased bacterial growth that leads to decreased AC and increased OS. Low levels of AC could be suggestive of increased OS and increased potential for oxidative damage [37].

Antioxidants, such as vitamin C and vitamin E, are found in the saliva and play an important role in the total antioxidative system of the oral cavity [38]. Salivary vitamin C concentration has been reported to be 6 to 10 μg/mL by Hegde et al. [39]. In concordance with this, in our study, mean AC levels in the saliva of pregnant and non-pregnant women were 6.59 μg/mL and 7.17 μg/mL of vitamin C equivalent, respectively. We did not measure vitamin C concentration directly in the saliva. However, on measuring ABTS radical scavenging activity vitamin C equivalent of saliva provides a good estimate, as the main AC effect is due to the vitamin C. Vitamin C helps to maintain the integrity of teeth and overall oral health by contributing to non-enzymatic antioxidant defense. Decreased serum and/or salivary vitamin C levels have been considered as one of the factors associated with dental caries [39]. We measured a 34% lower value of AC in the saliva of pregnant women compared to that of non-pregnant women. Therefore, decreased AC may predispose women to poor oral health and increase the risk of dental caries during pregnancy.

Excessive and uncontrolled production of ROS leads to OS, that in turn damages cellular structures and alters functions of DNA, protein, and lipids. Antioxidants and antioxidative system counteracts the ROS and/or prevent ROS formation [40,41,42]. Various kinds of ROS are regularly generated during cellular metabolic processes, and a low to moderate levels of ROS are physiological acting as signaling molecules [40,41]. MDA is one of the cellular lipid metabolites generated by the ROS reaction, and therefore, OS levels are generally expressed by the corresponding MDA metabolites concentration. Hence the increased level of OS is indicated by higher MDA levels. In this study, OS was found to be 16% higher in the saliva of pregnant women compared to non-pregnant women (*p* = 0.023).

Our study has some limitations. We did not measure NO in the serum samples parallelly with the saliva samples to avoid repeated invasive blood sampling from the pregnant study participants. However, salivary and serum NO levels are shown to have a positive correlation [43]. We compared healthy pregnancies from a general population with a selected group of non-pregnant women which mainly consisted of women working in the hospital and university and may possess better knowledge of oral health and oral hygiene. Additionally, our study population mainly consisted of White European women, and therefore the findings may not be directly applicable to other multi-ethnic populations. The oxidative stress and antioxidative capacity may depend on life-style factors including food intake and physical activity, which were not taken into account in this study. Similarly, salivary components might have changed due to some variation in its collection timing. Our study had adequate statistical power to demonstrate statistically significant differences in salivary OS levels between pregnant and non-pregnant women as well as differences between early second, late second and early third trimesters, among pregnant women. However, the desired sample size was not reached for the late third trimester due to increasing dropout of study participants with advancing gestational age. Although ABTS free radical assay and MDA measurement are well documented reliable assays, we have not performed other types of assays or measurements in serum or other body fluids from the same women to evaluate OS. This may limit the generalizability of our findings. Our study focused only on two caries-related bacteria, i.e., *SM* and *LB*, rather than investigating the whole oral microbiome. Although these are the most important pathogens, the possible role of other microbes in causing pregnancy associated changes in oral cavity cannot be ignored. Furthermore, we did not perform any clinical oral examination before saliva sampling. However, our study participants were healthy and none of them reported having any significant medical illness or oral health problems.

## 5. Conclusions

We report, for the first time, NO levels measured longitudinally in the saliva of pregnant women and demonstrate that salivary NO increases with advancing gestational age. OS and AC levels were stable during the second half of pregnancy. Normal pregnancy was associated with increased levels of NO and OS, and decreased antioxidant capacity compared with the non-pregnant state, and an abundant bacterial colonization of oral cavity by both *SM* and *LB* was observed among healthy pregnant women during second and third trimester of pregnancy indicating that pregnancy may have an adverse impact on oral health. Therefore, it is important to provide essential awareness and provision of optimal oral healthcare during pregnancy.

## Figures and Tables

**Figure 1 ijerph-18-09330-f001:**
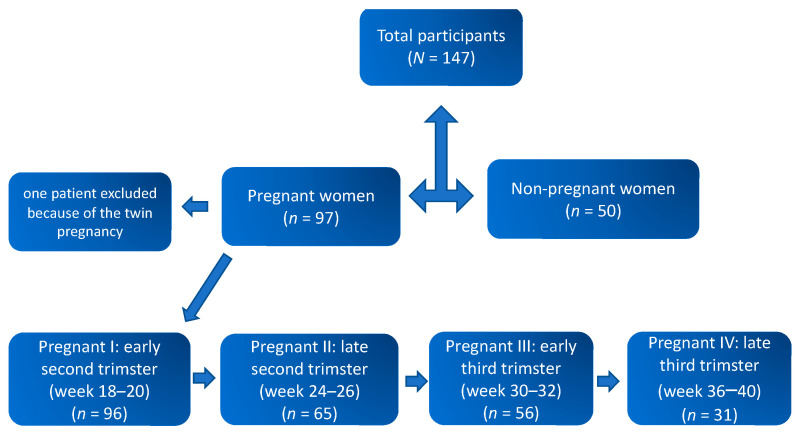
Chart illustrating the flow of study participants.

**Table 1 ijerph-18-09330-t001:** Nitric oxide (NO) equivalent nitrite levels (µM) in the saliva of pregnant women at different gestational ages and non-pregnant women.

Nitric Oxide (µM)	NP	Preg. I	Preg. II	Preg. III	Preg. IV
Mean	2.67	3.47	3.67	4.56	4.73
SD	1.55	2.49	3.04	4.23	2.87
MAX	7.58	13.49	16.89	22.69	13.97
MIN	0.52	0.20	0.19	0.43	1.41
*p*-value vs. NP *		0.0004	0.0002	<0.0001	<0.0001
*p*-value vs. Preg. *			0.4279	0.0005	<0.0001

Results are expressed as mean ± SD. * Bonferroni correction and Turkey’s posthoc test was used when multiple comparisons were performed. Preg. I: the early second trimester (18–20 weeks, *n* = 96), Preg. II: the late second trimester (24–26 weeks, *n* = 65), Preg. III: early third trimester (30–32 weeks, *n* = 56) and Preg. IV: late third trimester (36–40 weeks, *n* = 31), and NP: non-pregnant (*n* = 50).

**Table 2 ijerph-18-09330-t002:** Antioxidant capacity (AC) in the saliva of pregnant women at different gestational ages and non-pregnant women.

AC (%)	NP	Preg. I	Preg. II	Preg. III	Preg. IV
Mean (%)	80.82	60.35	60.80	58.40	60.50
SD	11.60	14.33	15.18	15.18	17.58
*p*-value vs. NP *	-	<0.0001	<0.0001	<0.0001	<0.0001
*p*-value vs. Preg. I *	-	-	0.8745	0.1972	0.8745

The results are shown as the percentage mean and SD values for each group expressed as ABTS radical scavenging capacity measured spectrophotometrically as optical density at 731 nm (OD_731_). * Bonferroni correction and Turkey’s posthoc test was used when multiple comparisons were performed Preg. I: the early second trimester (18–20 weeks, *n* = 96), Preg. II: the late second trimester (24–26 weeks, *n* = 65), Preg. III: early third trimester (30–32 weeks, *n* = 56) and Preg. IV: late third trimester (36–40 weeks, *n* = 31), and NP: non-pregnant (*n* = 50).

**Table 3 ijerph-18-09330-t003:** Oxidative stress level measured as MDA level in the saliva of pregnant women at different gestational ages and non-pregnant women.

MDA (nM)	NP	Preg. I	Preg. II	Preg. III	Preg. IV
Mean	0.85	0.93	0.95	0.97	0.96
SD	0.22	0.25	0.24	0.26	0.27
*p*-value vs. NP *	-	0.0083	0.0018	0.0005	0.0055
*p*-value vs. Preg. I *	-	-	0.4901	0.2017	0.4673

The results are shown as mean (SD) of malondialdehyde (MDA) nM values for pregnant and non-pregnant groups. * Bonferroni correction and Turkey’s posthoc test was used when multiple comparisons were performed. Preg. I: the early second trimester (18–20 weeks, *n* = 96), Preg. II: the late second trimester (24–26 weeks, *n* = 65), Preg. III: early third trimester (30–32.0 weeks, *n* = 56) and Preg. IV: late third trimester (36–40 weeks, *n* = 31), and NP: non-pregnant (*n* = 50).

**Table 4 ijerph-18-09330-t004:** *Streptococcus mutans* (*SM*) counts in the saliva of pregnant women at different gestational ages compared with non-pregnant women.

Group	Category 0 (%)	Category 1 (%)	Category 2 (%)	Category 3 (%)
Preg. I	11 (11.5)	25 (26.0)	35 (36.5)	25 (26.0)
Preg. II	6 (9.2)	22 (33.9)	24 (36.9)	13 (20.0)
Preg. III	6 (10.7)	13 (23.2)	22 (39.3)	15 (26.8)
Preg. IV	3 (10.0)	8 (26.7)	12 (40.0)	7 (23.3)
NP	20 (40.0)	12 (24.0)	11 (22.0)	7 (14.0)

Results are shown as number of subjects (%) in the categories 0 to 3. Preg. I: the early second trimester (18–20 weeks, *n* = 96), Preg. II: the late second trimester (24–26 weeks, *n* = 65), Preg. III: early third trimester (30–32 weeks, *n* = 56), Preg. IV: late third trimester (36–40 weeks, *n* = 31), and NP: non-pregnant (*n* = 50). Categories 0, 1, 2, and 3 represent the scoring for 0, <10^5^, 10^5^–10^6^, and >10^6^ CFU/mL SM bacterial colony forming unit (CFU) counts, respectively.

**Table 5 ijerph-18-09330-t005:** *Lactobacillus* (*LB*) counts in the saliva of pregnant women at different gestational ages compared with non-pregnant women.

Group	Category 0 (%)	Category 1 (%)	Category 2 (%)	Category 3 (%)	Category 4 (%)
Preg. I	23 (24.0)	41 (42.7)	16 (16.7)	9 (9.4)	7 (7.3)
Preg. II	11 (16.9)	21 (32.3)	20 (30.8)	4 (6.2)	9 (13.8)
Preg. III	5 (8.9)	24 (42.9)	13 (23.2)	5 (8.9)	9 (16.1)
Preg. IV	2 (6.7)	16 (53.3)	4 (13.3)	3 (10.0)	5 (16.7)
NP	15 (30.0)	15 (30.0)	11 (22.0)	7 (14.0)	2 (4.0)

Results are shown as number of subjects (%) in categories 0 to 4. Preg. I: the early second trimester (18–20 weeks, *n* = 96), Preg. II: the late second trimester (24–26 weeks, *n* = 65), Preg. III: early third trimester (30–32 weeks, *n* = 56), Preg. IV: late third trimester (36–40 weeks, *n* = 31), and NP: non-pregnant (*n* = 50). Categories 0, 1, 2, 3, and 4 represent the scoring for 0, 10^3^, 10^4^, 10^5^, and 10^6^ CFU/mL *LB* bacterial colony forming unit (CFU) counts, respectively.

## Data Availability

The data presented in this study are available, in anonymized form, on request from the corresponding author. The data are not publicly available due to privacy rules.

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
