# Peer review of "Nitric Oxide, Oxidative Stress and Streptococcus mutans and Lactobacillus Bacterial Loads in Saliva during the Different Stages of Pregnancy: A Longitudinal Study"

_ijerph, 2021, doi:10.3390/ijerph18179330_

Round 1

Reviewer 1 Report

The manuscript titled: “Oral inflammation, oxidative stress, and bacterial milieu during the different stages of pregnancies: a longitudinal study” shows the evaluation of NO levels, antioxidant capacity by ABTS, and determination of oral microbiota to assess differences in these parameters between pregnant and non-pregnant women. The manuscript is well written, scientifically-sound, and fits within the scope of IJERPH. I believe the major concern that should be addressed by the authors is justifying how ABTS and MDA are enough to quantify antioxidant capacity and oxidative stress in the samples. Minor revisions are suggested as follows:

  1. The abstract starts with nitric oxide, but NO is not reflected on the title. Please adjust the abstract or the title to include NO, as it seems NO is one of the major focuses of this research.
  2. Line 16: Please delete the space between “are” and “interconnected”.
  3. The Griess method, although extensively used to quantify “nitric oxide”, it is used to quantify nitrites. Please re-write and adjust it for the entire manuscript.
  4. Line 65: What does “nd” means?
  5. Lines 76-77: More studies are needed to claim that higher MDA levels and lower salivary antioxidant activity has been observed in patients with periodontitis.
  6. Line 85: Did the authors mean “some”?
  7. Line 136: Please provide the name and commercial information/manufacturer from the kit.
  8. Lines 153-154: Why did the authors quantified NO using tubes? Most Griess Reactions currently reported are adapted to microplates.
  9. Line 161: I do not think just quantifying ABTS is enough to indicate the assay was the total antioxidant capacity. Most authors reporting colorimetric antioxidant capacity methods usually use at least two different methods (e.g., DPPH, FRAP, CUPRAC, ORAC) to provide a better assessment of the antioxidant capacity. Can the authors provide a justification of just quantifying antioxidant capacity using 1 method?
  10. Line 179: As for ABTS, MDA alone is not enough to quantify oxidative stress. Please justify how just quantifying MDA would be enough to estimate oxidative stress.

Reviewer 2 Report

The aim of this study was to estimate the potential association between nitric oxide,  oral inflammation, oxidative stress OS, bacterial milieu at different pregnancy stages and healthy pregnancy.  This important period of life for women appears to have an adverse impact on oral bacterial milieu as demonstrated by significantly bacterial colonization, together with increased inflammation and OS levels, as well as decreased TAC levels.
The work is original and the research is well detailed. 

The authors sought to carry out a correct methodology, but naturally some questions raised. My considerations are:

Keywords: "saliva" should be added

Introduction is too long, please provide in one paragraph.
The sample size was not justified?
Were medical history, blood pressure or laboratory assessments evaluated specifically for the present study? Were the factors for the inclusion/exclusion criteria in the two groups also? It would be clear to add abovementioned criteria in table.

In limitations of the study it should be underlined that no diet control (any unfavorable one) or oral hygiene analysis were performed. During pregnancy usually number of meals and sugar intake are increased, bacterial colonization, oral pH may rise. Why only stimulated saliva was chosen for laboratory tests?

Conclusion paragraph is joined to the next part of the manuscript by some error propbably.

The purpose of the study is interesting and relevant in the context of dentistry.

Reviewer 3 Report

The paper shows that there are significant differences in salivary levels of NO, TAC and MDA between pregnant and non-pregnant women in a longitudinal study.

The article needs to be revised. First, a logic is unclear. Second, there are no data of important confounders. Third, the authors should follow the STROBE statement. Forth, there are many typos (L16, 28, 65, 70-88, Table 2, 394, 444, 445 and more). The current form is not acceptable.

TITLE

1) Please change the words, “oral inflammation” and “bacterial milieu” because the authors did not investigate oral inflammation but NO level only, and they did not investigate bacterial milieu but SM and LB levels only.

2) Please revise the title following the new logic.

ABSTRACT

1) Please revise the abstract following the new logic.

INTRODUCTION

1) The logic is unclear. First, why did the authors investigate only saliva samples without investigation of detail oral condition including DMFT and periodontal status? Why don’t they use serum sample that will not be affected by oral conditions? The reason that no study has performed is not appropriate. Second, NO level reflects one of the inflammatory status but not “oral inflammation”. SM and LB levels DO NOT reflect bacterial milieu. These are just dental caries related bacteria. Do the authors want to investigate the association between dental caries related bacteria and pregnancy? What I the mechanism? Finally, does salivary OS reflect pregnancy directly? Oral conditions can directly affect salivary OS. The authors need to reconsider the logic and show new one clearly.

2) Please add the study design of ref 9 and 10 (L61).

3) Please add the hypothesis before the aim following the STROBE checklist.

MATERIALS AND METHODS

1) The authors should follow the STROBE guideline. Some important factors of STROBE checklist are missed. Please add more in the text; design, period, bias, missing data, etc.

2) Please revise the design following the new logic.

3) Please move the actual number of participants to the result section.

4) Please add the detail data of validity, reliability and reproducibility, and examiner names in all measurements.

5) Please add the total time of salivary collection (? Minutes) and its timing (am ? to am?).

6) Please add the data of oral conditions (see above).

7) Please change the matching method based on the sample size estimation (see below).

8) If possible, please add the data after birth.

9) Please add the paired t-test. All statistical analyses need Bonferroni’s corrections.

RESULTS

1) Please add the flowchart following the checklist. Furthermore, please explain why the authors recruited the 97 and 50 participants even though the sample size estimation suggests 38 and 38 participants. Furthermore, how did the authors match the age? the number per group should 1:1 or 1:2 (the number of control group should be larger).

2) Please add the results based on new methods.

3) Please add the characteristic table following the STROBE checklist.

4) Please add the results of paired t-test (Preg 1 vs. others) and the name with Bonferroni’s correction in the Table 1-3. Please explain the decreased number of participants (97 to 31).

5) Please delete the Control in the Table 2.

6) Please add the test name and p-values.

DISCUSSION

1) Please revise the discussion and conclusion based on new results.

2) Please delete the paragraph about the strengthen because it is not appropriate.

3) Please add other comments about generalizability and more limitations.

Round 2

Reviewer 3 Report

The paper was not improved satisfactory and some questions are ignored.

The article needs to be revised. First, a logic is still unclear. Second, the authors do NOT revise the whole text according to my suggestions (There are still inappropriate words; inflammation, bacterial milieu, etc.). The current form is not acceptable.

ABSTRACT

  1. Please revise the abstract following the new logic.

INTRODUCTION

  1. The logic is still unclear. First, why don’t they use serum sample that will not be affected by oral conditions? The reason that no study has performed is not appropriate. Second, do the authors want to investigate the association between dental caries related bacteria and pregnancy? What is the mechanism? Do the authors want to suggest that stage of pregnancy can change the SM level? Finally, does salivary OS reflect pregnancy directly? Oral conditions can directly affect salivary OS.

If the authors hypothesized that the caries-related bacteria, OS level and anti-oxidant capacity in saliva change during pregnancy with advancing gestation, the authors should consider the mechanism how the stage of pregnancy affects the caries-related bacteria, OS level and anti-oxidant capacity in saliva. Then, they should state what we know now, why the caries-related bacteria, OS level and anti-oxidant capacity in saliva was only investigated and what the authors want to add in this study.

The authors did not add clinical data of oral conditions. They do NOT need the comments about periodontitis and poor oral health throughout the whole text.

The authors did not add the data of serum NO and OS levels. At least, they should add some comments and references that serum NO/OS level is associated with saliva one and oral condition has smaller effects on salivary NO/OS level than the stage of pregnancy. The readers will be confused about the causal relationship.

The authors need to reconsider the logic and show new one clearly.

This reviewer suggests a simple style because the current form has big limitations.

For example, the authors may state that focusing on NO and OS is an important aspect of pregnancy. Next, they may state what we know and what we don’t know. They may state that salivary levels of NO and OS are important and useful among pregnancy women and why they focus on the salivary levels. Then, they can hypothesize that NO level and OS in saliva change during pregnancy with advancing gestation. The aim of this study can be to investigate changes in salivary NO and OS levels during the development of pregnancy. In this case, the authors shoud delete the commnets about SM and LB.

  1. Please add the study design of ref 9 and 10 (L61) (reminder).

In the paper, the authors state, “There are a limited number of published studies measuring serum NO levels in women in relation to the pregnancy, but no study to our knowledge has performed NO measurements in the saliva of pregnant women longitudinally. “

The authors may say that there are a limited number of corss-sectional studies measuring serum NO levels in women in relation to the pregnancy [9, 10], but few logitudinal studies to our knowledge has performed NO measurements in the saliva of pregnant women.

  1. Please revise the hypothesis following the new logic.

 MATERIALS AND METHODS

1) The authors should follow the STROBE guideline (reminder). Some important factors of STROBE checklist are missed. Please add more in the text; period (from month, year to month, year), bias (we control XX bias by XX), missing data (we did not delete the data of participants with incomplete data and XX), etc.

2) Please revise the design following the new logic.

3) Please move the actual number of participants to the result section (reminder).

Please delete the numbers (97, 50, 65, 56, 31, and 50). Saliva samples were collected consecutively from 97 healthy pregnant women four times starting at 18-20 weeks to term pregnancy and from 50 healthy non-pregnant women. The samples were divided into 4 groups by gestational age; the first trimester (18 – 20 weeks, n = 97), the second trimester (24 – 26 weeks, n = 65), early third trimester (30 – 32.0 weeks, n = 56) and late third trimester (36 – 40 weeks, n = 31). A control group was comprised of healthy non-pregnant women of reproductive age (n = 50) without history of any chronic illness or recent disease.

4) The authors state that saliva samples were collected from participants in the daytime between 09:00-15:00. This is a limitation because salivary components are affected by time, especially morning and afternoon. Furthermore, the authors did not control diet and toothbrushing. These big limitations should be addressed.

5) The authors state that they need at least 38 individuals per group and recruited more participants to compensate for possible non-attendance or drop-outs at follow-up visits, inadequate sample collection or analysis failure, to ensure that there would be adequate number of samples/measurements for each stage of pregnancy. However, there are no comment about dropout rate and then final number of sample size. Please add these things.

Furthermore, the number of analyzed data in Preg. IV was 31, which is smaller than the size. The authors did not use matching methods. Please add the comments in the limitation.

7) The authors added the data on birth outcomes including mode of delivery, total blood loss during delivery; sex, birth weight and length of the neonates as well as 1 and 5 min Apgar scores are presented in the result section (Lines 234-239). If new logic includes that salivary NO/OS level predict birth outcomes, they should add the association between NO/OS level and birth outcomes.

RESULTS

1) Please add the flowchart following the checklist. Then, please explain why the number of participants reached 31.

2) Please add the results based on new methods.

DISCUSSION

1) Please revise the discussion and conclusion based on new results.

2) Please add other comments about limitations (see above).
